# Magnetic Induction Tomography: Separation of the Ill-Posed and Non-Linear Inverse Problem into a Series of Isolated and Less Demanding Subproblems

**DOI:** 10.3390/s23031059

**Published:** 2023-01-17

**Authors:** Tatiana Schledewitz, Martin Klein, Dirk Rueter

**Affiliations:** Institute of Measurement Engineering and Sensor Technology, University of Applied Sciences Ruhr West, D-45407 Mülheim an der Ruhr, Germany

**Keywords:** magnetic induction tomography (MIT), electromagnetic tomography, inverse problems, image reconstruction

## Abstract

Magnetic induction tomography (MIT) is based on remotely excited eddy currents inside a measurement object. The conductivity distribution shapes the eddies, and their secondary fields are detected and used to reconstruct the conductivities. While the forward problem from given conductivities to detected signals can be unambiguously simulated, the inverse problem from received signals back to searched conductivities is a non-linear ill-posed problem that compromises MIT and results in rather blurry imaging. An MIT inversion is commonly applied over the entire process (i.e., localized conductivities are directly determined from specific signal features), but this involves considerable computation. The present more theoretical work treats the inverse problem as a non-retroactive series of four individual subproblems, each one less difficult by itself. The decoupled tasks yield better insights and control and promote more efficient computation. The overall problem is divided into an ill-posed but linear problem for reconstructing eddy currents from given signals and a nonlinear but benign problem for reconstructing conductivities from given eddies. The separated approach is unsuitable for common and circular MIT designs, as it merely fits the data structure of a recently presented and planar 3D MIT realization for large biomedical phantoms. For this MIT scanner, in discretization, the number of unknown and independent eddy current elements reflects the number of ultimately searched conductivities. For clarity and better representation, representative 2D bodies are used here and measured at the depth of the 3D scanner. The overall difficulty is not substantially smaller or different than for 3D bodies. In summary, the linear problem from signals to eddies dominates the overall MIT performance.

## 1. Introduction

The diversity of requirements and applications for 3D imaging has produced an equally diverse range of techniques. One approach is magnetic induction tomography (MIT), which is attractive for several reasons, but which also involves considerable difficulties and limitations and has therefore remained in the research stage for a long time. In MIT, primary induction fields are projected into a measurement volume. Within an electrically conducting test body, eddy current fields are established and are shaped by the unknown conductivity distribution of the body. For example, the different conductivities of fatty and muscular tissue can be distinguished, and the internal and external geometry of simple objects can be recorded [1] with a linear scanning method, being one of the first MIT approaches. Secondary fields are detected by receivers [2,3], and their specific signals are used for computerized reconstruction of the searched conductivity distribution. Potentially advantageous features of MIT are its contactless, harmless, low-energy, fast, and low-cost operation. In addition, electrical conductivity is an alternative contrast method to X-ray or ultrasound. In biomedicine, MIT has been repeatedly promoted for detecting features in deep areas of the human body [4]. MIT could be used, for example, in the non-invasive mapping of cardiac tissue conductivity [5]. Other biomedical applications arise in the field of rapid, non-hazardous and non-contact lung diagnostics [6] or in the rapid detection of internal bleeding or tumors in the human skull [7,8].

Nevertheless, MIT is more likely to serve as a supplement than a replacement for established imaging methods because of the inherently blurred nature of the induction fields that act as a well-permeating and probing “radiation”. The spatial resolution and sensitivity over distance (i.e., body interior) is modest. Relevant inhomogeneities can thus be difficult to localize. The forward problem (i.e., the received signals as a function of known or assumed conductivities) can be unambiguously computed; however, the inverse problem for obtaining the searched conductivities from measurements is both ill posed and nonlinear. Small inaccuracies in a practical measurement can cause large deviations or instabilities in the solution. The overall task is often even underdetermined because the number of distinguishable conductivity voxels can readily become higher than the number of independent measurement data [9].

## 2. Settings

In the biomedical regimen, with typical conductivities σ of biological tissues from 0 S/m to 1 S/m [10,11,12], the sensitivity of frequently published and annular MIT designs is sufficient near the surface but decreases considerably towards the central interior [9,13,14] (Figure 1a), which then becomes hardly detectable [15,16]. The issue originates from disadvantageous eddy current topologies in the conducting volumes, where a vortex core with vanishing eddies typically tends to center itself in the body. In order to gain more control here, a planar MIT geometry [17,18] combined with a linear scanning method [1] has become the basis of our research activities.

The previously presented and practically realized 3D MIT scanner [19,20] (Figure 1b) used in this study deviates considerably from the circular setups Figure 1a, as it has an increased sensitivity or relatively increased eddies for the central interior of test bodies or persons. The large MIT scanner comprises only one extended and periodically arranged excitation coil—an undulator—for projection of a steady state and spatially periodic excitation field at 1.5 MHz. A suitable undulator periodicity corresponding to a certain spatial frequency of the field topology, and a distinct suppression of lower spatial frequencies were shown to excite comparably more eddies in the body center (Figure 2) than was achieved with the more common and single excitation loops, where lower spatial frequencies are included in the projected fields. The lower components dominate over distance and excite unfavorable eddy topologies, almost regardless of the orientation of the ordinary excitation loop (Figure 1a).

For the scanner considered here, six opposing and gradiometrically aligned butterfly receivers provide six measurement values from the body. More 3D data are obtained by a mechanical scan procedure. The test object or person moves linearly past the gap between excitation and receivers within some seconds, and six quasi-continuous signal traces result as a function of the scanning position. Each signal has 256 entries, rather than only one number. Although a mechanical scan procedure might be viewed as unfavorable for accurate signaling, viable data with high finesse and low noise were actually obtained: the relevant signal of the test object itself could be resolved to about 0.1% (60 dB useful dynamic range). The undulator improved the accessibility of the depth of the volume [19,20], and considerably improved the center signals (+26 dB) with respect to a setting with more unfavorable and circular excitation coils (similar to Figure 1a). For the first time, practical measurements allowed 3D reconstructions throughout the depth of low conducting/low contrast and voluminous test bodies, with sizes and electrical properties similar to those of a human torso.

In technical realization, a mathematically ideal and quasi-infinite undulator plane was replaced with a large and finite undulator with 11 current stripes. The gradiometric receivers suppressed the directly incident primary signals. The dominant portion of the signals is provided by the interesting eddy currents inside the object (or, more precisely, their secondary fields) [21]. However, due to the limited bandwidth of the signal traces (the adjoint receiver fields undergo a depression of the high spatial frequencies over distance), the independent data actually obtained amounts to fewer than 200 readings per receiver. Nevertheless, the data is sufficed to reconstruct a blurred conductivity distribution.

The received signal is calculated in the forward problem as the integration of the inner product of the adjoint induction field of the receivers (represented by the vector potential $A_{R}$) and the eddy current density field J in the object [19,22]. $A_{R}$ is constant and has to be (pre-)calculated only once [19]. By contrast, the J in the entire object must be recalculated for each scan position $x_{s}$, which then leads to a considerable computational effort [23]. An iterative algorithm has typically been previously used for MIT inversion, as well as for this MIT scanner [19,20]. The approach was based on a sensitivity matrix (Jacobian) and a regularized Landweber inversion [24,25]. The Landweber method is a variant of the steepest gradient descent linear iterative method widely used in optimization theory to solve ill-posed inverse problems. Only the first-order derivative is used, which requires many iterations before reaching a minimum. Another disadvantage is semi-convergence. The image error rapidly decreases in the beginning, but increases again after reaching a minimum point. The optimal number of iterations can be determined from the previous information about the capacity measurements, but usually a fixed number of iterations (1000 in this work, with zeros as the initial value) is chosen empirically [26].

The 3D conductivity reconstruction was earlier directly determined from the signal of a body with included perturbations (i.e., the elements in the Jacobian describe signal changes as a result of a locally altered conductance). Differential measurements were used for practical 3D MIT reconstructions (i.e., the difference of a body with included and “unknown” perturbations and a known and similar body without perturbations). The practical differential signals matched the numerically calculated differential signals from the forward problem well [20], supporting the applied forward theory. Conversely, and due to imperfections of the practical setup, the total or overall signals revealed noticeable deviations (about 5%) between practice and theory, and this, in addition to other issues, prevented absolute reconstructions. The numerical setup of a Jacobian is computer intensive, as the many individual signal entries for each local deviation in the body require multiple forward calculations. The Jacobian must even be repeatedly recalculated, as the overall nonlinear inversion is iteratively approached. Moreover, the process does not allow for simple and instructive insights, as deriving handy conclusions for favorable measures is difficult from this type of integral strategy. This is even truer for the MIT inversions using machine learning, which was also studied for this particular MIT setup [27]. On the one hand, machine learning can provide fast and good solutions for ill-posed inverse problems in imaging [28]. Emerging applications of artificial intelligence in healthcare range from one-dimensional bio-signal analysis to prediction of medical events such as seizures, cardiac arrest, diagnosis, etc. [29]. On the other hand, there are also specific drawbacks, such as prolonged and computer-intensive training runs, or the difficult analysis or interpretation of deep learning models in terms of actually measurable results or possible improvements for the technique [30]. In particular, scenarios that deviate significantly from the training and occur acutely can cause difficulties.

The more theoretical topic presented here is the separation of the MIT inversion into a series of decoupled and less demanding subproblems, as schematically illustrated in Figure 3. The interfaces of the individual processes roughly follow the physical interfaces of the forward problem. The four separated tasks can be better controlled in detail and allow clearer insights into issues such as possible error sources, instabilities, or potential improvements. Remarkably, the overall computer efficiency is also accelerated by about one order of magnitude because the subproblems can be processed more directly. For example, a repeated and costly setup of the previously acclaimed [19,20] Jacobian becomes unnecessary here.

The sinusoidally periodic excitation field exploited in the present method allows the J to be obtained for every scan position $x_{s}$ by suitable superposition of only two J components from distinct positions before the undulator [19] (Figure 2b,c), named here as ϕ (Phi) and ψ (Psi). During a scan procedure, the components $J_{\phi }$ and $J_{\psi }$ sinusoidally alternate. As indicated in Figure 2a, the symmetric position ϕ indicates that the traveling object is centered in front of an exciter copper strip, and the antisymmetric position ψ indicates the position between the two strips [19]. The offset between ϕ and ψ amounts to a quarter period of the undulator. Also, other pairs can be arbitrarily utilized as a basis for J by shifting the two reference positions with an offset γ to ϕ+γ and ψ+γ. A reasonable γ would extend from zero to a quarter of the undulator period, with everything subsequently in periodic repetition. The only two required J fields provide a complete basis for all occurring current fields along $x_{s}$. The number of unknown J elements becomes considerably smaller than considering 200 different J fields for a scan in a non-periodic excitation field from, for example, a single and localized primary coil.

Notably, the only two describing eddies $J_{\phi }$ and $J_{\psi }$ (Figure 2b,c) are essential ingredients for the proposed strategy: in numerical discretization, the number of independent current elements (only about 50% of all current elements in Figure 2b,c are independent) for the two J simulates the number of unknown conductivities in the body. By contrast, for classical MIT (Figure 1a) (for example, with 8 or 16 different J from individually controlled excitation coils), the total number of eddy elements considerably exceeds the number of unknown conductivities. This could be one reason why this separation of the inversion has not been considered so far.

The first subproblem is a suitable splitting (Figure 3a) of the received signal S into two components, $S_{\phi }$ and $S_{\psi }$, which would result from the two individual and alternating eddy current fields, $J_{\phi }$ and $J_{\psi }$. This task is already a linear and somewhat ill-posed inverse problem, as $S_{\phi }$ and $S_{\psi }$ together carry more independent data or information than the practically available signal S. However, often quite satisfactory results are obtained from exploiting basic sampling theorems for bandwidth-limited signals (resulting from blurred receiver fields) and from additional restrictions.

In the second task, the two separated signals $S_{\phi }$ and $S_{\psi }$ are used to reconstruct two corresponding fields of magnetization, $M_{\phi }$ and $M_{\psi }$, inside the body; these can be considered as the origin of the projected (secondary) magnetic induction field. This step is also a linear and ill-posed inverse problem that is tackled here with the Landweber method. In numerical computing, the continuous M is replaced by arrays of discrete magnetic dipole moments $m_{\phi }$ and $m_{\psi }$. The m essentially represents a decomposition of the interconnected eddy current field into independent and local elementary vortices.

The intermediate magnetization, m, is applied for several reasons, rather than directly reconstructing the eddies from the signals S. The number of searched current elements (they are not all independent) is significantly higher than the number of m required to fully describe the eddy current field. Eddy elements, when considered as a whole, have to fulfil the node rule, which is not automatically ensured for current fields directly obtained from an inversion of the signals. Fortunately, the number of $m_{\phi }$ and $m_{\psi }$ elements together simulates the number of ultimately searched and discretized conductivities.

The transition from magnetization to eddy currents is made by a curl operation, a principally lossless, quick, and linear step. The curl of M results in an eddy current field J, which is the exclusive source of the magnetization here. A numerical curl from the discrete $m_{\phi }$ and $m_{\psi }$ roughly doubles the number of discrete (and not independent) eddy current elements. In this respect, m represents a vector potential for the current elements, as a continuous body magnetization M can be a vector potential for J.

In the fourth and final step, the searched conductivities are reconstructed from the two representative current fields at their base positions ϕ and ψ. This operation represents the inversion of an eddy current calculation from a given conductivity distribution and a known excitation field. The problem is non-linear; however, it is fairly benign and appears not to be a substantial impairment for the MIT method, as the currently presented and iterative algorithm (Figure 3b) reasonably converges. More concise mathematical formulations and an even more rigid and efficient solution of this task may be reported in the near future from this group.

All steps (except the third) of the inversion introduce inaccuracies or losses to the reconstruction; therefore, the MIT inversion remains ill-posed. When assuming an overall resulting inaccuracy as a relative number (e.g., a percentage), applying a differential MIT strategy makes sense. Only the smaller differential signal between the unknown test object and a similar, well-known, and predefined or estimated object is reconstructed or, better yet, used to readjust the predefined object. Although the smaller correction carries almost the same level of relative inaccuracy, the absolutely introduced inaccuracy with respect to the whole body decreases. The initial estimation of a suitable reference object, at present, is a typical task for machine learning. In addition to the total signal of the body, the outer shape (optically available) and many typical cases and inner patterns from a database can also be used. However, as body estimation with machine learning is not the topic here, we restrict ourselves for now to a reference object with the same contour and only homogeneous conductivity as the simplest and but often helpful assumption. Nevertheless, a better initial estimation will ultimately provide smaller resulting losses and inaccuracies due to small or even (in best case) vanishing differential signals.

The right elements in Figure 3a were applied for differential MIT based on a first body estimation. The elements on the left side show the non-retroactive inversion chain. In addition to the technical quality and information content of the measurement signals and the advantage of estimating a well-fitting reference body in differential MIT, the first and particularly the second task of the chain (i.e., from received signals back to body magnetization) appear to limit the overall performance of the MIT. These are graspable and linear inverse problems, and more direct hints can be derived for better performance.

## 3. Materials and Methods

In this study, we examine only a 2D test sheet with dimensions 40 cm × 40 cm, discretized into squares 2 cm × 2 cm in size. In contrast to 3D bodies, the advantage of an only 2D sheet is that it provides a much more instructive representation and interpretation of vector fields and underlying effects, as already exploited in [20] to illustrate the principal support from the undulating excitation field. Notably, the characteristic difficulties of 3D MIT are conserved due to the geometric distance of the 2D sheet to both the excitation and the receiving plane. Both the excitation and the receiving field are blurred over the distance in z-direction (Figure 2a); therefore, sharp features in the 2D sheet ultimately become unresolvable, which is related to the issues for features in the depth of 3D bodies.

In a general 3D situation, an eddy current density field J is induced inside the body, where J is a function of the body’s conductivity distribution σ and the exciting induction field, which may be described by an exciting vector potential $A_{E}$. In the typical numerical calculation required, the continuous σ and J are discretized into voxels or lumped elements (Figure 4). J is condensed into currents I along the directed length elements Δl of the discrete network, which also applies in 2D. The conductivity σ is condensed into discrete resistors along those length elements Δl, the interconnected grid of resistors forms a passive, linear, spatially extended, and exclusively resistive network in 2D or in 3D. Contributions from reactive elements (capacitors or inductors in the grid) are neglected here because, in our biomedical approach and at sufficiently low operation frequencies ω=2πf of the excitation field, neither capacitive displacement currents (ωϵ≪σ) nor inductive reactance ($\omega \mu d^{2}\ll \frac{1}{\sigma }$) substantially alter the resistively determined J within the characteristic depth d of a body in a 3D scenario. The phase relation between the exciting currents in the undulator and the eddy currents in the body then approaches 90°, as well established by experimental observations from test persons or voluminous saline phantoms with conductivity in the biomedical range.

To obtain currents I in a network of resistors, the voltage sources must somehow be introduced. From the exciting primary field $A_{E}$, the induced voltage U in each length element Δl of the representative grid is given by:(1)$$U=-j\omega \left(A_{E}\cdot \Delta l\right),$$
where $A_{E}$ applies to the center position of Δl and jω represents the time derivation at angular frequency ω. The 90° phase shift is introduced by the imaginary unit j. The directions of $A_{E}$ and Δl are considered with the inner product of these vectors.

With all given U and all given resistors of the network, the multiple I can be unambiguously calculated (Figure 4b). Although this statement is intuitively clear (“Ohm’s law”), the computational effort for many repeated I calculations (e.g., for the Jacobian) can become significant for larger networks (e.g., those with thousands of nodes). Large systems of linear equations then have to be solved. The essential difficulty of this task is the determination of the resulting electrical scalar potential in the grid via nodal analysis. This is already well established and shall not be further discussed here.

An important and underlying assumption here is that the excitation field $A_{E}$ is somehow given and fixed (i.e., it is virtually not altered by the multiple I with their additional [secondary] field). This issue is addressed further below.

Six gradiometrically aligned receivers detect the secondary induction field, which is projected from the multiple current elements I or, more generally, from the continuous J of a 3D body. The calculation of secondary induction fields from the J and their impact on the receiver loops, however, would be laborious. This can be relieved by the reciprocity theorem [22,31]: the receiver signal S from a body under the regime of weak coupling [32] can be calculated by the inner products between the adjoint (and merely fixed) vector potential $A_{R}$ of the receiver and the eddy current density J over the body volume Ω:(2)$$S=\overset{\;}{\underset{\Omega }{\int }}J\cdot A_{R}d\Omega$$

The so-called weak coupling [15] regime is required for (2) and supports less laborious calculations of the eddy currents from the given conductances and a given and fixed primary field $A_{E}$. Weak coupling can be practically realized with sufficiently low operation frequencies (say <2 MHz for the human body). This means that both the primary and the secondary induction field are barely absorbed or deformed by the low conducting volume (always keeping a 3D situation in mind). Both fixed fields $A_{E}$ and $A_{R}$ can therefore be pre-computed and do not add extra effort during MIT computation. Weak coupling is quite the opposite of inductive probing (eddy current testing) of well-conducting metals, where the exciting field undergoes considerable deformation and ultimately annihilation at an already shallow skin depth due to the relatively much stronger eddy currents. Here, the virtually stiff primary $A_{E}$ simplifies the calculation of the eddies inside a volume (1), and the virtually undistorted secondary from the eddies allows exploitation of the fixed $A_{R}$ and reciprocity in (2).

In numerical discretization (Figure 4b), the received signal can be calculated from the currents I along the belonging and directed conductor length (or voxel size) Δl of the grid. Again, the $A_{R}$ is applied to the center position of the Δl. The summation over all voxels p approaches the integral in (2), and the expression also applies to 2D grids, where i is the voxel number:(3)$$S=\overset{p}{\underset{i=1}{\sum }}I_{i}A_{R_{i}}\cdot \Delta l_{i}$$

Current densities J are not definable in a 2D sheet; however, in a 3D situation and for voxel volume v with current densities J, the transition between (2) and (3) is
(4)$$I\Delta l=\overset{\;}{\underset{v}{\int }}Jdv$$

### 3.1. Magnetic Dipole Moments as a Basis for Eddy Current Fields

A material with internal currents is magnetically active. For an isolated and non-ferromagnetic body (relative permeability $\mu_{r}=1$; magnetic moments or spins on atomic scale are averaged out as well), the describing magnetization M of the material represents the vector potential of eddy currents J inside the body (5). The magnetization M becomes unambiguous for the necessary condition M=0  in the surrounding vacuum or air, where J=0.
(5)J=∇×M

In discretization and for a voxel size v, a magnetic momentum m of a voxel (Figure 4c) is given by
(6)$$m=\overset{\;}{\underset{v}{\int }}Mdv$$

From (4) and (5), it directly follows that:(7)$$I\Delta l=\overset{\;}{\underset{v}{\int }}\nabla \times Mdv$$

We exploit for the discretized grid (then use difference quotients for the curl operation, with characteristic distances Δl instead of the mathematical differential, shown further below)
(8)$$I\Delta l=\nabla \times \overset{\;}{\underset{v}{\int }}Mdv=\nabla \times m$$
with the flux density B=∇×A, the expression (2) can be rewritten as:(9)$$S=\overset{\;}{\underset{\Omega }{\int }}M\cdot B_{R}d\Omega$$

For a numerical discretization with p elements, this translates to:(10)$$S=\overset{p}{\underset{i=1}{\sum }}m_{i}\cdot B_{R_{i}}$$

The support for (9), (10) and results from
(11)$$S=\overset{\;}{\underset{\Omega }{\int }}\left(\nabla \times M\right)\cdot A_{R}d\Omega =\overset{\;}{\underset{\Omega }{\int }}M\cdot (\nabla \times A_{R})d\Omega$$
and
(12)$$S=\overset{p}{\underset{i=1}{\sum }}\left(\nabla \times m_{i}\right)\cdot A_{R_{i}}=\overset{p}{\underset{i=1}{\sum }}m_{i}\cdot (\nabla \times A_{R_{i}})$$

The identities in the triple products (11) and (12) can be geometrically argued from a necessarily constant amount of a 3D volume expanded by three vectors, regardless of their order. Here, in (11) and (12), the vector product inside the brackets provides the normal vector of a base area, and the inner product with a third vector, “height,” then results in a volume. The identity for (12) is also numerically obtained in Figure 5 below.

In the MIT inversion approach used here (Figure 3a), the linear Equation (10) serves to reconstruct multiple m from a signal S, whereas Equation (8) readily delivers eddy currents I in the conductor elements Δl from the m.

As already mentioned, a direct determination of the current elements IΔl from the signal S is obstructed by two issues. On the one hand, the number of unknown current elements (= number *k* of interconnected Δl in the grid Figure 4) is significantly greater than the number of actually independent currents because the number *n* of grid nodes also introduces additional restrictions from the node rule. The smaller number of actually independent currents in the grid is represented by the right term of the following inequality:(13)k>k−n+1

This equals the required number of describing moments m, which is almost intuitively obtained in 2D (Figure 4c).

On the other hand, as ensured by (8), all the current elements from the vector potential m will automatically establish an eddy current field without sources and sinks. ∑I=0 ( or in continuum: ∇·J=0) is thus implicitly fulfilled for every node in the network.

In the other direction, from IΔl to m, the integration (or for discretized elements: summation) is also directly available, given the condition that both I and m are zero outside the body.

Figure 5 shows arbitrary examples for the calculated signals in one receiver, following the expressions (14) and (15), as described further below. An absolute signal from one 2D object is shown with perturbations, together with a differential signal for two slightly deviating objects (i.e., with and without the introduced perturbation). For the examples, two entirely different calculation paths (12) lead to virtually equal signals; that is, (3) with the adjoint receiver vector potential $A_{R}$ and eddy current elements I and (9) with the adjoint receiver flux density $B_{R}$ and moments m. Furthermore, the signal bandwidth in Figure 5 clearly does not fully exploit the more than 200 measurement samples of a scan procedure. The signals appear low-passed, caused by the blurred field $A_{R}$ (or $B_{R}$) of a distant receiver. Over a distance, sharp and localized features (e.g., edges) cannot provide sharp responses in the signal, and filigree features readily become undetectable below a practical noise floor. Basic sampling theorems (e.g., the Shannon theorem) can show that the signals in Figure 4 actually do not provide more practically useful data than about 30 independent measurement values, at best.

### 3.2. Inversion Chain

#### 3.2.1. Signal Splitting

For any arbitrary scan position $x_{s}$ of the unknown body, the momentary eddy current can be expressed based on $J_{\Phi }$ and $J_{\Psi }$ as:(14)$$J_{x_{s}}=J_{\Phi }\cos\left(\frac{2\pi x_{s}}{D}\right)+J_{\Psi }\sin\left(\frac{2\pi x_{s}}{D}\right)$$
where D is the distance between two conductors of undulator.

The two signal components can be numerically calculated (in MATLAB, for example) according to the following expressions, which are based on (12):(15)$$S_{\Phi }\left(x_{s}\right)=\cos\left(\frac{2\pi \Delta x}{D}\right)\overset{p}{\underset{i=1}{\sum }}I_{\Phi_{i}}\left(x,y,z\right)\left(\Delta l_{i}\left(x,y,z\right)\cdot A_{R_{i}}\left(x+x_{s},y+y_{0},z+z_{0}\right)\right),$$
(16)$$S_{\Psi }\left(x_{s}\right)=\sin\left(\frac{2\pi \Delta x}{D}\right)\overset{p}{\underset{i=1}{\sum }}I_{\Psi_{i}}\left(x,y,z\right)\left(\Delta l_{i}\left(x,y,z\right)\cdot A_{R_{i}}\left(x+x_{s},y+y_{0},z+z_{0}\right)\right)$$

$A_{R}\left(x+x_{s},y+y_{0},z+z_{0}\right)$ is the incident vector potential in the coordinate system of the body at a shifted x-position $x_{s}$ ($y_{0}$, $z_{0}$ = const, see Figure 1b and Figure 2a).

Only the sum is an actually available measurement signal:(17)$$S=S_{\phi }+S_{\psi }$$

As for the inversion of (17), an advantage of the typically rather bandwidth-limited signal S is that it is already fairly successful at splitting by sampling into distinct ϕ and ψ positions (Figure 5b,c) or into intentionally shifted positions ϕ+γ and ψ+γ. For these defined positions, one of the eddy current fields is set to 100% active, while the other is definitely zeroed (14–16) (i.e., isolated components from $S_{\phi }$ and $S_{\psi }$ are available). The sparse samples from all ϕ and ψ positions already allow for a first estimation $S_{\phi E1}$ and $S_{\psi E1}$ for the interesting and continuous $S_{\phi }$ and $S_{\psi }$. Further corrections for $S_{\phi E1}$ and $S_{\psi E1}$ in intermediate sampling positions (between the dashed vertical lines in Figure 5b,c, where both $S_{\phi }$ and $S_{\psi }$ contribute in unknown portions) are obtained from the trivial statement $S=S_{\phi }+S_{\psi }$. The error from $S_{\phi E1}+S_{\psi E1}\neq S$ is added as weighted parts to the two components to obtain a second estimation $S_{\phi E2}$ and $S_{\psi E2}$, which fulfills $S=S_{\phi E2}+S_{\psi E2}$ for all scan positions and better approaches the unknown and searched $S_{\phi }$ and $S_{\psi }$. For the weighted correction, a helpful step is to (automatically) choose a shift γ in a way that the energies of the preliminary obtained $S_{\phi +\gamma ,E1}$ and $S_{\psi +\gamma ,E1}$ are as different as possible. In one of the best cases, the energy of one component becomes virtually zero (e.g., for $S_{\phi +\gamma ,E1}$), so the correction for intermediate samples is applied to 100% for $S_{\psi +\gamma ,E1}$. In this benign case, the finally resulting $S_{\psi +\gamma ,E2}$ exactly equals the searched $S_{\psi +\gamma }$ as well as S, whereas $S_{\phi +\gamma ,E2}$ is virtually zero here. Interestingly, this particular situation would apply to a rather good body estimation in differential MIT, where only one and local perturbation is still “hidden” somewhere in the unknown body of interest.

An even more trivial splitting without losses would be possible for signals with a more limited bandwidth, which can regularly be obtained from more distant receivers. The bandwidth (or better, the highest signal frequency) can then become smaller than the halved sampling frequency for safe contributions of $S_{\phi }$ and $S_{\psi }$, and a lossless signal splitting is then possible in any case. However, the latter measure with more distant receivers is not desirable because we strive for as much information as possible from the depth of the measurement space.

#### 3.2.2. Inversion Chain, from Signals to Moments

The magnetic dipole moments for the local perturbations in two positions are reconstructed using the iterative Landweber method. This process provides an energy minimized (pseudo-)inversion of a forward description. Here, a forward describing matrix (Figure 6) r×u is created, where r is the dimension of the receiver signal length and u is the dimension of the number of moments or squares to which the weakly conducting body is discretized.

As the moments in the sheet can only extend in the z-direction (Figure 4c), the forward matrix in Figure 6 with standardized moments essentially displays the z-components of the receiver flux density in various positions. The forward matrix is calculated once and is valid for both $S_{\phi }$ and $S_{\psi }$. For the inversion from given signals toward moments, the Landweber iteration provides a reasonable moment distribution, even in underdetermined situations (e.g., fewer independent signal data are available than interesting moment positions). In Figure 6, the signal vector with 1200 entries appears to overdetermine only 400 moments. Actually, the opposite applies here, as the bandwidth is rather limited and 6 signals provide, in total, fewer than 200 independent measurements. With basic signal processing, the signal vector in Figure 6 can be compressed to fewer than 200 entries without loss of information regarding the MIT task (Figure 6d), i.e., fewer than 200 components are actually independent. The underdetermined problem of moving from signals back to moments becomes unambiguous due to the additional incorporation of energy minimization (Landweber), which makes the resulting 400 moments not entirely independent (which they were in the original forward description, however). The discretization with 20 × 20 moments is therefore finer than the signaling can resolve. As a result of, for example, Shannon’s theorem, any even finer discretization could therefore not yield signals that could not also be produced by the 20 × 20 discretization.

#### 3.2.3. Inversion Chain, from Moments to Eddy Current Elements

The curl operation J=∇×M (5) for a discretized 3D lattice (8) becomes, via differences and finite lengths:(18)$$I_{x}\Delta l_{x}\left(x,y,z\right)=\frac{m_{z}\left(x,y+0.5\Delta l_{y},z\right)-m_{z}\left(x,y-0.5\Delta l_{y},z\right)}{\Delta l_{y}}-\frac{m_{y}\left(x,y,z+0.5\Delta l_{z}\right)-m_{y}\left(x,y,z-0.5\Delta l_{z}\right)}{\Delta l_{z}}$$
(19)$$I_{y}\Delta l_{y}\left(x,y,z\right)=\frac{m_{x}\left(x,y,z+0.5\Delta l_{z}\right)-m_{x}\left(x,y,z-0.5\Delta l_{z}\right)}{\Delta l_{z}}-\frac{m_{z}\left(x+0.5\Delta l_{x},y,z\right)-m_{z}\left(x-0.5\Delta l_{x},y,z\right)}{\Delta l_{x}}$$
(20)$$I_{z}\Delta l_{z}\left(x,y,z\right)=\frac{m_{y}\left(x+0.5\Delta l_{x},y,z\right)-m_{y}\left(x-0.5\Delta l_{x},y,z\right)}{\Delta l_{x}}-\frac{m_{x}\left(x,y+0.5\Delta l_{y},z\right)-m_{x}\left(x,y-0.5\Delta l_{y},z\right)}{\Delta l_{y}}$$

The same statement but intuitively better understandable, he separated and localized current circulations in Figure 4c (blue circles around the red m) need to be merely superimposed or summed to receive the total and interconnected eddy current field in Figure 4b. Thereby, only a difference between two neighboring m would result in a net current between them.

All m and I are defined or set to zero outside the body volume (3D) or area (2D), as the empty space is magnetically passive and is also not conducting. For a cubic lattice with $\left|\Delta l_{x}\right|=\left|\Delta l_{y}\right|=\left|\Delta l_{z}\right|$ the expression simplifies somewhat. As already mentioned, all resulting I automatically fulfill the node rule for arbitrary m arrays (i.e., the obtained I definitely establish an eddy field inside the body (∇·J=0)).

#### 3.2.4. Inversion Chain, from Known Eddy Currents toward the Conductivity Field

This process is non-linear and tackled iteratively here with the rather stable algorithm schematically outlined in Figure 3b. A prerequisite is an initial conductivity estimation, which could be the same as the body estimation for differential MIT measurements in Figure 3a. However, the quality of the initial estimation is less critical here. A simple estimation is a homogeneous body with same outer contour. For this, the eddy currents $J_{\Psi_{E}}$, $J_{\Phi_{E}}$ of the initially estimated body in the positions Ψ and Φ are calculated in the forward problem. Conversely, the actually measured eddy currents $J_{\Psi_{M}}$, $J_{\Phi_{M}}$ are available from the previous task, and these currents typically will deviate from the first estimation. The initial and all subsequently estimated conductivity distributions are then iteratively corrected by the following procedure:

In the first step, the signs of the measured and estimated current elements are omitted. The current amounts from both the unknown and the estimated body are set for each voxel or area element (pixel) in the Ψ and Φ positions, respectively;The respective fields with current amounts from positions Ψ and Φ are further condensed into an array of totally measured current amounts $I_{M}$ of the unknown and the known (or estimated) body $I_{E}$:(21)$$I_{M,E}=\sqrt{I_{\Psi_{M,E}}{}^{2}+I_{\Phi_{M,E}}{}^{2}}$$The quotient of $I_{M}$ and $I_{E}$ serves as an array with correction factors K for each conductivity voxel in 3D (or pixel in 2D) reconstructions:(22)$$K=\frac{I_{M}}{I_{E}}$$

For a measured amount $I_{M}$ higher than an estimated $I_{E}$, the correction K becomes >1, and vice versa;

The estimated conductivity array is multiplied elementwise with the K array (i.e., whenever the measured and local current intensity is higher than the current of the estimation, the local conductivity is increased, and vice versa). The new conductivity cannot become smaller than zero. In addition, an upper conductivity limitation to “1” is also applied in the code. This is justified by the knowledge that a biomedical conductivity typically cannot exceed 1 S/m;The corrected conductivity distribution leads, after a new forward calculation, to a new set of estimated eddy currents $I_{E}$, which regularly are more similar to the measured currents $I_{M}$. Thus, the new conductivity estimation better approaches the unknown conductivities.

After some repetitions (typ. < 10) of the procedure, the conductivity estimation approaches the unknown object. The values stabilize or do not further improve when the repeatedly corrected current estimation $I_{E}$ virtually equals the measured current intensity $I_{M}$; all entries in the K array then approach 1. Notably, the overall process appears rather stable, as reasonable eddy current reconstructions (although these are blurred in MIT) deliver potentially reasonable and non-oscillatory conductivity solutions, even in underdetermined situations. Other highly deviating and even more exact approaches for this task will be presented from this group in the near future. We are satisfied with the preliminary results, as the performance of this iterative and nonlinear inversion from currents to conductivities does not appear to be the dominant issue for the overall MIT performance.

## 4. Results

This section presents basic results of the reconstruction chain (Figure 3). Note, that the goal here was not a further optimization of the used and very simple receiver geometries (only simple conductors here), but it is rather about the principle of the inversion process. Differential MIT is applied and a homogeneous sheet with the same outer contour serves as a first estimation.

As described in Section 3.2.1, the two alternating eddy currents $J_{\Psi }$ and $\;J_{\Phi }$ allow a splitting of the differential signal into two components $\;S_{\phi }$ and $S_{\psi }$. The two differential signals are inverted to two arrays of differential magnetic dipole moments (Section 3.2.2).

Figure 7 shows the reconstruction of differential magnetic dipole moments (Figure 7b,c) in both positions Ψ and Φ from differential signals of the “unknown” object in Figure 7a. The subsequent step is the conversion from differential moments to differential currents (18)–(20)—a lossless step (Figure 7d,e). Following the process map in Figure 3a, the differential current fields are added to the absolute current fields of the initially estimated and homogeneous sheet (Figure 2b,c), resulting in the reconstructed current fields (Figure 7f,g), which deviate somewhat from the initial estimation Figure 2b,c.

In the next step, the nonlinear conductivity reconstruction from the current fields is performed. The input variables are fields of the actually measured (Figure 7f,g and the resulting Figure 8a) and estimated current intensities (21) (Figure 2b,c, resulting to Figure 8b), which serve as corrections (22) for the estimated conductivity distribution.

Figure 9 shows iterative conductivity reconstructions of the unknown object, based on the intensity in Figure 8a. The likely locations and the character of the perturbations settle after about 5 iterations.

Additionally introduced and artificial noise in the signals results to only moderately affected reconstructions (Figure 9d). The here applied noise level is 40 dB SNR in the differential signals, which is comparable to practically observed noise for the MIT scanner [19].

In comparison to the unknown original in Figure 9a, the apparent imperfections and artifacts in Figure 9c mainly originate from the previous inversion processes (particularly the ill-posed and linear Landweber inversion from signals to moments), and more iterations in this nonlinear process will not improve the outcome.

The becomes more obvious in Figure 10, where different distances (in the z-direction, see Figure 1b and Figure 2a) are applied between the sheet and simple receivers. More distant receivers result in more blurred fields $B_{R}$, which directly affect the sharpness of the reconstructed moments and the subsequent current fields, finally blurring the resulting conductivity fields. For comparison, Figure 10a shows a reconstruction with perfectly determined currents, where the imperfections of the last and non-linear inversion step evolve. Perfect current measurements can theoretically be achieved with receivers at zero distance (i.e., a contacting measurement). However, the task in MIT is a contactless measurement over distance, particularly inside the depth of an unknown body.

Interestingly, the iterative reconstructions from ideally measured or known eddy currents to unknown conductivities perform better in 3D (Figure 11), although 3D is not the general topic here. One reasonable explanation for this better outcome with respect to Figure 10a is that environmental current distortions or deviations around local perturbations are smaller in 3D, due to a more rapid decay or relaxation of field distortions over a distance in 3D than in 2D. The field topology of the measured current intensities $I_{M}$ in 3D is therefore more similar to the conductivity map right from the beginning (i.e., the initial K-Array in Figure 11b equals the conductivity estimation after the first correction).

The conductivity reconstruction of low conducting and voluminous bodies can apparently become very close to the unknown original if the signal splitting and the reconstruction of magnetic moments would ideally perform.

## 5. Discussion and Conclusions

This theoretical work follows a rather practical demonstration [19] and describes an MIT inversion procedure that is separated into a series of non-retroactive steps. The tasks can be treated individually and are each less demanding on their own. Compared to previous strategies over the entire inverse problem, the presented inversion chain is computer efficient (10 s in a standard laptop for the whole procedure, instead of minutes), as no particularly laborious operations, such as setting up a Jacobian, are required. Briefly available prior information (e.g., a special outer contour or safely suspected inhomogeneities) about the examination objects can be readily introduced into the process (Figure 12) and does not lead to a prolonged computation, or has to be defined beforehand. Compared to machine learning methods, no prolonged training time is required for completely unexpected circumstances or bodies [30].

Despite the promising initial results, there is still room for optimization. The linear sampling method proved to be suitable, for example, for solving inverse problems in related research fields [33].

The method can be applied to a recently realized 3D MIT scanner [19,20,27], where essentially only two sinusoidally alternating eddy current fields, ***J_φ_*** and ***J_ψ_***, provide the basis for all signals *S*. The approach appears less suitable for commonly published and circular MIT setups with multiple excitation coils (Figure 1a) and more eddy current fields, where the number of independent and unknown moments can considerably exceed the number of searched conductivities. The non-linear reconstruction from known eddy current fields to unknown body conductivities (***J_φ_***, ***J_ψ_*** → σ) is now recognized as comparably well-performing, particularly in 3D.

More issues appear in the linear processes of MIT inversion: dominant losses result in the reconstruction of current fields from bandwidth limited signals (*S_φ_*, *S_ψ_* → ***J_φ_***, ***J_ψ_***) and further losses can originate from signal splitting (*S* → *S_φ_*, *S_ψ_*). The latter procedure is not particularly analyzed in the present contribution, as the splitting losses, in many cases, are definitely small or even zero. More generally, the correlation between splitting results and the actually unknown signal components approaches a value of more than 90% in most cases. In differential MIT, the targeted small signal differences already have a practical noise floor or uncertainty within the relevant bandwidth. This can readily extend to or exceed several % and is already of a similar order as the splitting losses.

The main losses result from *S* → m, as the rather limited signal bandwidth of the inductive measurements entirely inhibits a reconstruction of small or sharp m features with higher spatial frequencies. In our examples, and depending on the receiver distance (Figure 10), only about 10% of the possible spatial bandwidth in the shown arrays is available from the inherently low passed signals; 90% of structural information (smaller or more granular features, sharp edges) is irrevocably lost. Only additional information or prior knowledge from other sources (e.g., already introduced in the initial body estimation) could introduce finer structures (Figure 12). Note that the initially estimated object and its currents can extend over the full bandwidth; this presumed information is added without losses at the bottom of Figure 3a.

As the difficulties presented by only linear and isolated problems are better to analyze and handle, specific adaptions (e.g., of the geometries of the undulator as well as of the receivers) can lead to an improved situation when considering practical system noise and other imperfections. Inductor geometries and their resulting field topologies can gradually improve the signals and the useful signal bandwidth. The measures have already proven to substantially influence the overall MIT performance. In addition, and as a matter of principle, the depth of the measuring chamber in the z-direction is decisive for the signal quality (see the influence of the z-distance in Figure 10). For a vanishing z-distance to the currents (Figure 10a), almost all problems are ultimately resolved. A 3D MIT, however, enforces a certain z-depth of the measurement space, and this thickness should be restricted as much as possible.

Before refining the inductor geometries for modifying and improving the principal signal quality, however, the entire inversion chain must also be presented for 3D scenarios (e.g., moments must be reconstructed in three spatial directions, instead of only one normal z-direction in 2D, as utilized here). As the B-field of the exciting undulator in Figure 1 has essentially only x- and z-components, the magnetization and the resulting m of a 3D-body can also be expected to point to a majority in the x- and z-direction. Besides taping all voxels with $m_{x}$ and $m_{z}$, a single layer of $m_{y}$ (presumably in the horizontal middle plane of the body) will then suffice to describe every possible eddy current field in the 3D body (i.e., the total number of m in x-, y- and z-directions then equals the right term in (13), which is the required number of independent current elements for a discretized volume).

Approaches to 3D scenarios following Figure 3 are currently being investigated. The principles are not fundamentally different, and since successful 3D reconstructions have already been demonstrated with other algorithms using this scanner [19,20,27], the structural information is definitely present in the signals. However, in addition to the greater error-proneness of a 3D code, the difficult visualization of 3D vector fields also makes the tracking of possible errors more difficult. Nevertheless, the nonlinear step “reconstruction of conductivities from known currents” can already be considered as well be performed (Figure 11). The remaining overall problem is essentially linear.

## Figures and Tables

**Figure 1 sensors-23-01059-f001:**
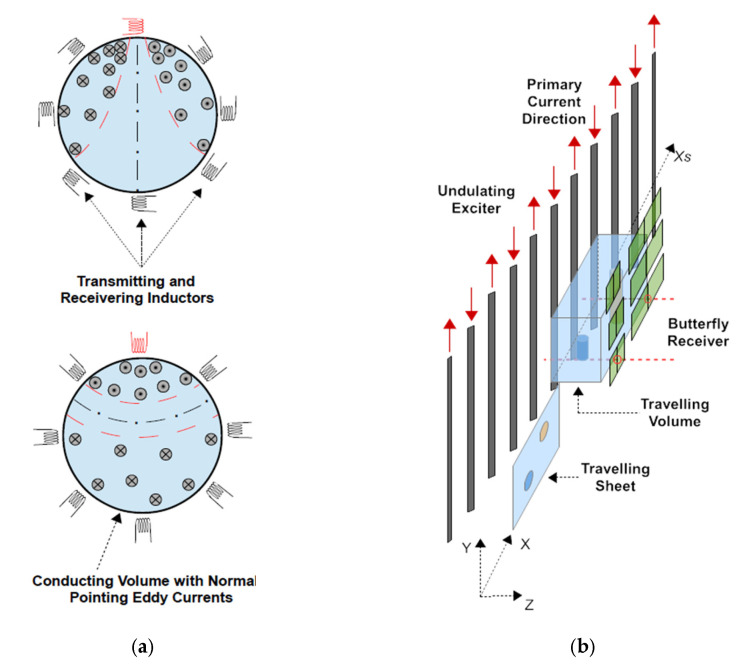
Structures of different MIT systems: (**a**) Top view of a typical annular MIT setup with small and localized transducers in two different orientations. The highest current densities appear near the exciting inductor (red) and a central region with vanishing eddy currents occurs [20]; (**b**) Illustration of a planar MIT scanner with a large and undulating exciter (undulator).

**Figure 2 sensors-23-01059-f002:**
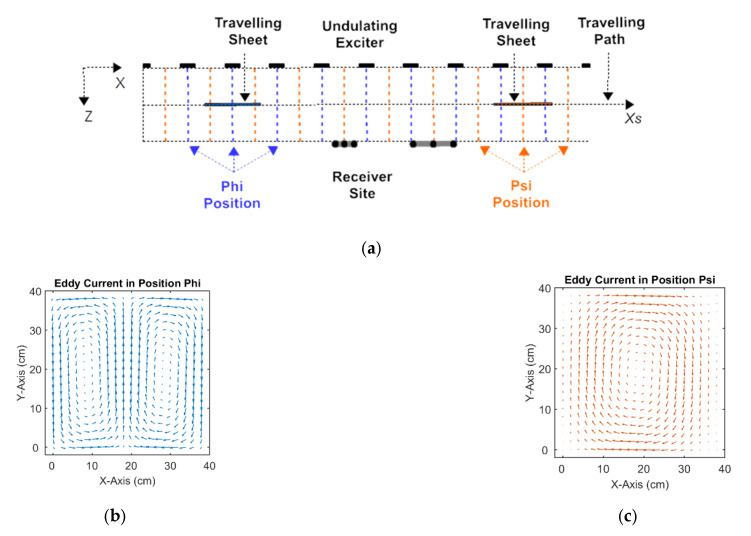
Planar MIT system and excited eddy currents: (**a**) Top view. The traveling 3D body or 2D test sheet moves in the X direction through the measuring range between the undulator and butterfly receiver array; (**b**) Homogenous 2D sheet in scan position Phi and eddy currents. Higher current densities in the center region are also preserved for 3D bodies; (**c**) Homogeneously conducting 2D sheet in scan position Psi.

**Figure 3 sensors-23-01059-f003:**
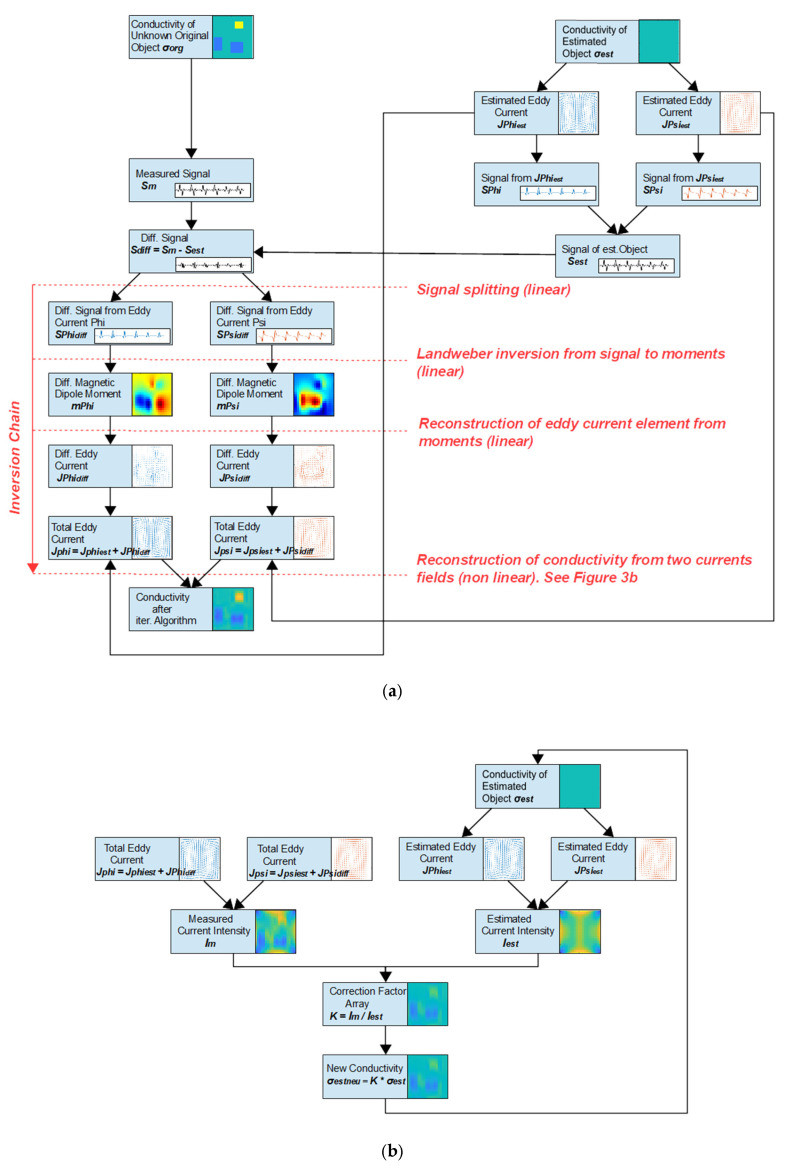
Process schematics. Proposed separation of the MIT inversion chain into a series of non-retroactive, less-demanding, and more insightful subproblems (red descriptions): (**a**) The overall inversion from a measurement (left column) of an unknown object can be relieved with differential MIT (right column): a suitable body estimation can be used to obtain smaller differential signals, which ultimately result in a relatively reduced number of errors and deviations; (**b**) The inversion task from known current fields toward conductivities is a nonlinear and benign task, which is tackled here with an iterative procedure.

**Figure 4 sensors-23-01059-f004:**
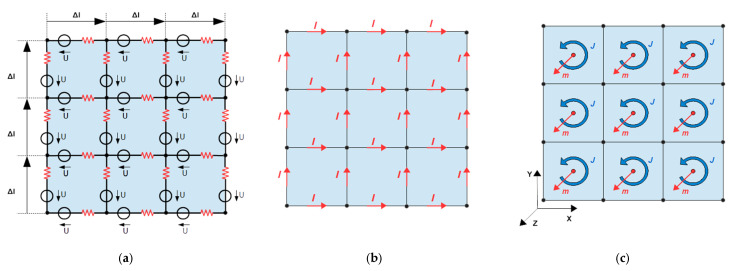
The discretized 2D object: (**a**) Representative electrical network with induced voltages; (**b**) The resulting eddy current field is represented by discrete current elements; (**c**) Magnetic moments point in the z-direction and represent localized and isolated current circulations, i.e., the eddy field (**b**) is decomposed into independent elements. Superposition of all elementary vortices results in the global eddy field in (**b**). Note that the number of sufficiently describing moments (here, 9) is smaller than the number of non-independent currents (here, 24) in (**b**).

**Figure 5 sensors-23-01059-f005:**
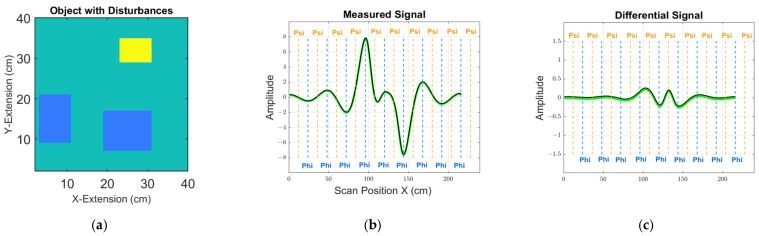
Measured object with perturbations and simulated receiver signals. The signals are calculated by the dot product of the currents and vector potentials (green line) and by the dot product of the magnetic dipole moments and flux densities (black line, small offset applied for better visibility). Vertical and dashed lines indicate the positions Phi (blue) and Psi (orange); (**a**) Measured object with perturbations; (**b**) Total signals with disturbances; (**c**) Differential signals calculated by the difference between a total signal with and without perturbations.

**Figure 6 sensors-23-01059-f006:**
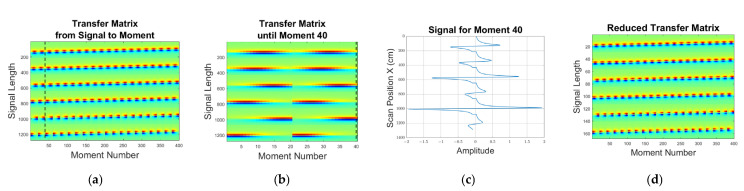
The transfer matrix from magnetic dipole moments to received signals. In vertical rows, showing the overall received signal from 6 receivers, each individual receiver signal extends over 200 samples. The 20 × 20 sheet hosts 400 independent moments (horizontal axis). Positive signal deflections are indicated in red, negative deflections in blue, and green indicates no deflection or zero: (**a**) The total transfer matrix from normalized moments to received signals, a signal collection for each position. The black line indicates the 40th moment; (**b**) Magnified transfer matrix until the 40th moment; (**c**) The signal sequence from 6 receivers for the 40th moment (dashed line in (**a**,**b**); all other moments are zero for this entry); (**d**) Due to sampling theorems, the transfer matrix can be condensed to less than 200 signal entries without losses.

**Figure 7 sensors-23-01059-f007:**
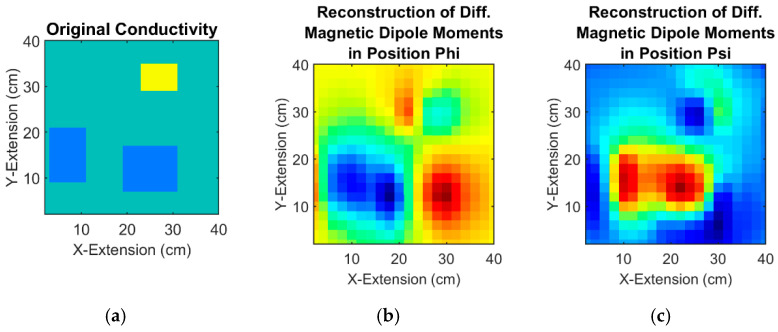
Reconstruction of the differential magnetic dipoles and resulting eddy currents from differential measurement of an inhomogeneous 2D object: (**a**) “unknown” object. Green background 0.5 S/m, higher conductivity 1 S/m in yellow, lower conductivity 0.25 S/m in blue; (**b**) show the reconstruction of the differential magnetic dipole moments using the differential signals in position Phi; (**c**) the differential magnetic dipole moments in position Psi; (**d**) Reconstruction of the differential eddy current in position Phi from moments; (**e**) Differential eddy current in position Psi. (**f**) Calculated total eddy current in position Phi from the differential eddy current in position Phi and the estimated eddy current in position Phi; (**g**) Total eddy current in position Psi.

**Figure 8 sensors-23-01059-f008:**
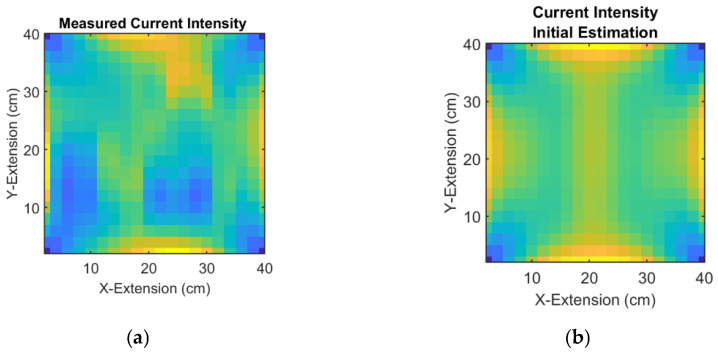
Total current intensity. Yellow color indicates high current intensity, dark blue indicates zero currents: (**a**) The total current intensity $I_{M}$ from measurements of the unknown and inhomogeneous sheet (Figure 7a); (**b**) Total current intensity $I_{E}$ of a homogeneous sheet, which serves as the first estimation here.

**Figure 9 sensors-23-01059-f009:**
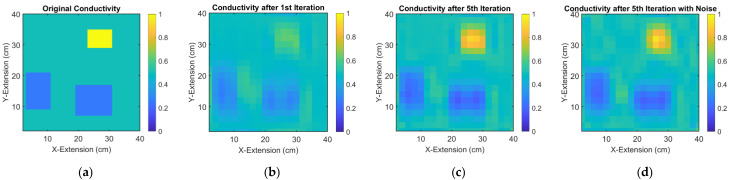
Conductivity reconstruction of a 2D, weakly conductive, and inhomogeneous object from measured currents: (**a**) original measurement object; (**b**) 2D reconstruction of the object after 1 iteration; (**c**) 2D reconstruction of the object after 5 iterations; (**d**) 2D reconstruction of the object after 5 iterations with artificial noise in the signal (40 dB SNR).

**Figure 10 sensors-23-01059-f010:**

Reconstructions of a weakly conductive and inhomogeneous 2D object, showing the influence of the z-distance between object and receivers: (**a**) Currents are determined by contacting measurement. The only source of imperfections is the non-linear inversion from currents to conductivities; (**b**) Conductivity after the 5th iteration with a 5 cm distance between the measuring body and receiver; (**c**) Conductivity after the 5th iteration with a 10 cm distance between the measuring body and receiver; (**d**) Conductivity after the 5th iteration with a 15 cm distance between the measuring body and receiver.

**Figure 11 sensors-23-01059-f011:**
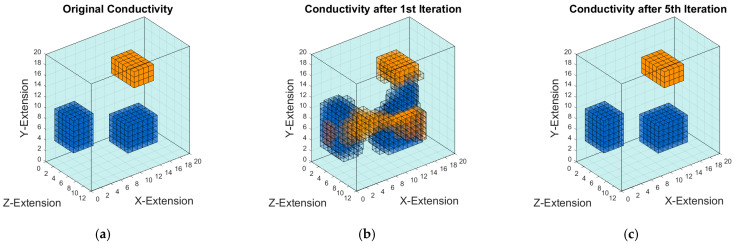
Conductivity reconstruction of a weakly conductive and inhomogeneous 3D body from ideally known eddy currents. The orange color indicates the conductivity 1 S/m. Blue objects indicate 0.25 S/m conductivity, background 0.5 S/m: (**a**) Original object with disturbances; (**b**) 3D reconstruction after the 1st iteration; (**c**) 3D reconstruction after the 5th iteration.

**Figure 12 sensors-23-01059-f012:**
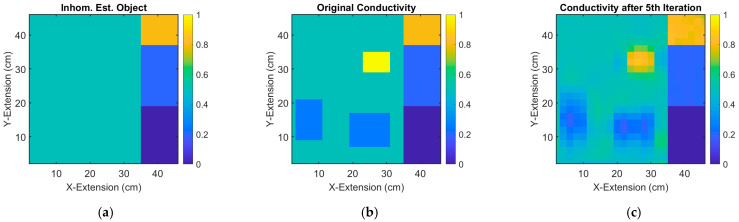
Conductivity reconstruction of a 2D, weakly conductive, and inhomogeneous object 46 cm × 46 cm: (**a**) Previously estimated object with pronounced inhomogeneities on the right side (prior knowledge for differential MIT). (**b**) Actual object with additional and unknown deviations. (**c**) 2D reconstruction. The previously suspected features are only confirmed by the MIT measurement, but no need to be lossy reconstructed. On the other hand, the unexpected deviations are revealed with the losses typical for MIT.

## Data Availability

Not applicable.

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
