# Peer review of "Magnetic Induction Tomography: Separation of the Ill-Posed and Non-Linear Inverse Problem into a Series of Isolated and Less Demanding Subproblems"

_sensors, 2023, doi:10.3390/s23031059_

Round 1

Reviewer 1 Report

Please see attached report

Reviewer 2 Report

The work is required to highlight the improvement based on analysis of this research. However, the analysis part is missing.

Reviewer 3 Report

This thesis focuses on Magnetic Induction Tomography: (MIT) carrying out a non-linear inverse problem study with creativity. But some issues need to be considered, as follows in the attachment

Round 2

Reviewer 1 Report

The authors addressed all my concerns in this new version. Nonetheless, a discussion on the reference [2-in my previous review] has not been implemented in the manuscript, although it is strongly related to the current study. Furthermore, it is based on shape gradient descent optimization.  The Landweber method, on which this manuscript is based, is a variant of the standard steepest gradient descent linear iterative method.  As the author mentioned it is "widely used in optimization theory to solve ill-posed inverse problems" 

It is then important in my view to discuss the pros and cons of the current work in regard to other work to showcase the current strength and competitivity. 

Overall, I keep this comment to the authors while proofreading, as I appreciate the substantial work done so far. 

Reviewer 2 Report

The manuscript is now ready for publication

Reviewer 3 Report

This manuscript has been revised according to the comments. I recommend accepting the manuscript.